# Culture of Cancer Cells at Physiological Oxygen Levels Affects Gene Expression in a Cell-Type Specific Manner

**DOI:** 10.3390/biom12111684

**Published:** 2022-11-14

**Authors:** Ricardo Alva, Fereshteh Moradi, Ping Liang, Jeffrey A. Stuart

**Affiliations:** 1Department of Biological Sciences, Brock University, St. Catharines, ON L2S 3A1, Canada; 2Centre for Biotechnology, Brock University, St. Catharines, ON L2S 3A1, Canada

**Keywords:** oxygen, physioxia, hyperoxia, cell culture, cancer cells, transcriptomics, differential gene expression, hypoxia-inducible factor, HIF-2α, mtDNA-encoded genes

## Abstract

Standard cell culture is routinely performed at supraphysiological oxygen levels (~18% O_2_). Conversely, O_2_ levels in most mammalian tissues range from 1–6% (physioxia). Such hyperoxic conditions in cell culture can alter reactive oxygen species (ROS) production, metabolism, mitochondrial networks, and response to drugs and hormones. The aim of this study was to investigate the transcriptional response to different O_2_ levels and determine whether it is similar across cell lines, or cell line-specific. Using RNA-seq, we performed differential gene expression and functional enrichment analyses in four human cancer cell lines, LNCaP, Huh-7, PC-3, and SH-SY5Y cultured at either 5% or 18% O_2_ for 14 days. We found that O_2_ levels affected transcript abundance of thousands of genes, with the affected genes having little overlap between cell lines. Functional enrichment analysis also revealed different processes and pathways being affected by O_2_ in each cell line. Interestingly, most of the top differentially expressed genes are involved in cancer biology, which highlights the importance of O_2_ levels in cancer cell research. Further, we observed several hypoxia-inducible factor (HIF) targets, HIF-2α targets particularly, upregulated at 5% O_2_, consistent with a role for HIFs in physioxia. O_2_ levels also differentially induced the transcription of mitochondria-encoded genes in most cell lines. Finally, by comparing our transcriptomic data from LNCaP and PC-3 with datasets from the Prostate Cancer Transcriptome Atlas, a correlation between genes upregulated at 5% O_2_ in LNCaP cells and the in vivo prostate cancer transcriptome was found. We conclude that the transcriptional response to O_2_ over the range from 5–18% is robust and highly cell-type specific. This latter finding indicates that the effects of O_2_ levels are difficult to predict and thus highlights the importance of regulating O_2_ in cell culture.

## 1. Introduction

Mammalian cell culture is often used to study cell physiology in health and disease. For this purpose, environmental parameters such as temperature and pH are regulated to recreate as closely as possible the in vivo conditions. However, while cells in most mammalian tissues are exposed to 1–6% oxygen in vivo [1], cell culture is routinely performed in incubators that regulate CO_2_ but not O_2_. At sea level, atmospheric O_2_ levels are ~21%, and headspace O_2_ in conventional incubators thus equilibrates to ~18% O_2_ due to high humidity and the addition of 5% CO_2_. Despite being referred to as ‘normoxia’, 18% O_2_ is substantially hyperoxic relative to the cellular microenvironment in vivo. Increasing evidence indicates that the hyperoxic conditions of cell culture affect multiple biological processes, including reactive oxygen species (ROS) production [2], redox homeostasis [3], proliferation and differentiation [4], bioenergetics [5], mitochondrial network dynamics [5], and response to drugs [6] and hormones [7]. These effects of non-physiologically high O_2_ levels can compromise the ability of cell culture models to recapitulate in vivo disease pathophysiology (reviewed in [8]).

Cancer cell lines are widely used in basic research to study cancer pathophysiology; however, they are routinely cultured in 18% O_2_. To gain a better understanding of the effects of this supraphysiological O_2_ levels in cell culture we studied the transcriptional response of four cancer cell lines to O_2_ levels in standard cell culture (18% O_2_) versus physioxia (5% O_2_). While physioxia is a range (typically 1–6% O_2_) rather than a set value, we chose 5% O_2_ for this study as we have accumulated data from other studies using this value [2,5,7,9], which is typical of many mammalian tissues in vivo [1]. Furthermore, one challenge of maintaining physioxia in vitro is avoiding pericellular hypoxia due to the lower gradient for O_2_ entry into media. We have previously shown that culture at 5% O_2_ in the incubator headspace maintains pericellular O_2_ in media at above 3.7% O_2_ under conditions used here [10].

We used RNA-seq and bioinformatic approaches to analyze differential gene expression of four human cancer cell lines cultured for 14 days at either 5% or 18% O_2_. The cell lines used in this project were LNCaP (prostate adenocarcinoma), Huh-7 (hepatocellular carcinoma), PC-3 (prostate adenocarcinoma), and SH-SY5Y (neuroblastoma). We chose these cell lines because we have previously measured the effects of O_2_ on a wide range of cellular activities in them, including energy metabolism, mitochondrial dynamics, ROS production, and response to drugs such as resveratrol [5,9]. Inclusion of SH-SY5Y cells was done to increase the breadth of cell type. Further, these cell lines are not only widely used in cancer research, but also as surrogates of primary cells to study their physiology and pathology. For example, SH-SY5Y cells are used in models of neurodegenerative disease [11] and ischemia/reperfusion models [12] and Huh-7 are frequently utilized to study hepatic xenobiotic metabolism [13] and fatty liver disease [14]. Using this approach, we asked whether the transcriptomes of all four cell lines were sensitive to O_2_ levels in the range from 5% to 18% O_2_, and whether the effects of O_2_ were similar amongst cell lines, or cell-line specific. We found that the effects of O_2_ on the transcriptomes of these cell lines were substantial and largely cell-line specific. Nonetheless, we identified interesting patterns in gene expression among these cell lines, such as a differential but strong induction of genes encoded by mitochondrial DNA (mtDNA) and upregulation of HIF-1/2 targets in physioxia.

## 2. Materials and Methods

### 2.1. Cell Culture

LNCaP, SH-SY5Y, Huh-7, and PC-3 cell lines were purchased from ATCC (Manassas, VA, USA). Cell passages between 15–19 were used throughout this study. Upon thawing, cells were cultured in 10-cm plates with Plasmax (Ximbio, London, UK) supplemented with 2.5% FBS and 1% penicillin/streptomycin (Sigma-Aldrich; St. Louis, MO, USA) for a week in a humidified 5% CO_2_ (~18% O_2_) to allow for acclimatization to Plasmax (physiological cell culture medium; see [15]) and reduced FBS concentration. Afterwards, cell lines were incubated in a humidified 5% CO_2_ incubator at either 5% or 18% O_2_. Three replicates per each cell line were used in each condition. For the experimental groups kept at 5% O_2_, Plasmax media was preincubated overnight in the 5% O_2_ incubator to allow for gas equilibration. Sub-culture was performed with 0.25% Trypsin-EDTA (Sigma-Aldrich, St. Louis, MO, USA) every 3 or 4 days, when cells reached ~80% confluence. Media was refreshed every 24 h and cells were routinely monitored for mycoplasma contamination. Cell culture at either 5% O_2_ or 18% O_2_ was performed for 14 days. Cells were seeded at a density of 2 × 10^6^ cells/plate prior to RNA extraction.

### 2.2. RNA Isolation

Total RNA was extracted using the RNeasy Plus Mini Kit (QIAGEN, Toronto, ON, Canada) according to the manufacturer’s instructions. RNA integrity was assessed using 1.5% agarose gel electrophoresis, while RNA concentration and purity were evaluated as A260/280 ratio using a Thermo Fisher Scientific Nanodrop spectrophotometer. RNA samples were snap frozen in liquid nitrogen and stored at −80 °C until being sent to Novogene (Sacramento, CA, USA) for sequencing and analysis.

### 2.3. Sequencing and Differential Gene Expression Analysis

Quality check (QC), library preparation, sequencing, and differential expression analysis were performed by Novogene. Paired-end at 150 bp (PE150) high throughput Illumina sequencing was performed at a sequencing depth of 40 million reads per sample. Reads were aligned to the *Homo sapiens* reference genome (GRCh38) using Hisat2 v2.0.5 [16]. Gene expression levels were estimated by calculating FPKM (fragments per kilobase of transcript per million mapped sequence reads), which were further adjusted by edgeR program package [17] through one scaling normalized factor. Differential expression analysis was performed using the edgeR R package. A *p*-value < 0.05 and |log_2_FC|≥ 1 were set as threshold for significantly differential expression, as done previously [10].

### 2.4. Functional Enrichment Analysis

Prior to functional enrichment analysis, the list of differentially expressed genes (DEGs) was further reduced to genes with a Benjamini adjusted *p*-value (*p*_adj_) < 0.1 and a FPKM ≥ 1 in at least one of the experimental groups in order to produce a concise list of enrichment terms which reflect the most strongly affected genes. Functional enrichment analysis was performed using the Database for Annotation, Visualization, and Integrated Discovery (DAVID) [18]. Enriched Gene Ontology (GO) terms, Kyoto Encyclopedia of Genes and Genomes (KEGG) pathways, and Reactome pathways were selected for functional annotation. A raw *p*-value and Benjamini adjusted *p*-value (*p*_adj_) of 0.05 were applied for identifying the most statistically significant enriched annotation terms.

### 2.5. Correlation Analysis of PC-3 and LNCaP Data with Gene Expression Data from the Prostate Cancer Transcriptome Atlas (PCTA)

Data from the Prostate Cancer Transcriptome Atlas (PCTA) was downloaded from [19], containing a dataset of expression levels of 18,390 genes from 2115 human prostate tumor samples. Upon removing genes with FPKM = 0 in any of the experimental groups, our lists of DEGs upregulated at 5% O_2_ and 18% O_2_ in both LNCaP and PC-3 were independently cross-referenced with the PCTA dataset to find overlapping genes. We then analyzed the correlation between the change in expression levels (log_2_FC) in our samples and the PCTA dataset (mean log_2_FC from the 2115 samples) by performing a Pearson correlation test with the software GraphPad Prism 8.

## 3. Results

### 3.1. Oxygen Levels in Culture Strongly Modulated Transcript Abundance Cell-Line Specifically

The abundance of over a thousand transcripts were affected by O_2_ in each cell line. In general, more differentially expressed genes (DEGs) showed higher expression at 5% O_2_ than at 18% in all cell lines. In addition, there was substantial variation between the four cell lines in their sensitivities to O_2_ (Figure 1A). For example, 2126 DEGs were identified in LNCaP cells, including 433 upregulated at 18% O_2_ and 1693 at 5%. In contrast, SH-SY5Y was shown to be the least sensitive to O_2_ among the cell lines, with only 386 transcripts upregulated at 18% O_2_ and 848 at 5%. The full lists of DEGs for LNCaP, Huh-7, PC-3, and SH-SY5Y cells are available in Appendix A, respectively.

A remarkable result was the extremely limited overlap between cell lines in terms of the identities of the DEGs (Figure 1B). Only four genes were identified as being O_2_-sensitive in all four cell lines. Of these, *BOLA2B* was the only protein-coding gene. The BolA protein family has important roles in Fe–S cluster biogenesis, iron and Fe–S cluster trafficking and storage, and iron sensing and regulation [20]. Differential expression of *BOLA2B* may thus have an impact in redox sensing and signaling. Interestingly, *BOLA2B* was found to be upregulated at 5% O_2_ in all cell lines, except Huh-7, where it was upregulated at 18% O_2_. Even amongst the two prostate cancer cell lines, LNCaP and PC-3, where 2126 and 1461 genes were differentially expressed, respectively, only 192 were shared between both cell lines. Similarly, of the 2099 transcripts affected by O_2_ in Huh-7 cells, 1638 (78%) were exclusively affected in this cell line. This indicates that O_2_ effects on gene expression are highly specific to a given cell line. This in turn makes it difficult to predict how the non-physiological O_2_ levels of standard cell culture are affecting cell biology in general terms.

Functional enrichment analysis revealed that different biological processes and pathways were enriched by O_2_ level in the four cell lines (Figure 2). For example, the most significantly affected pathway in LNCaP cells was TGF-β signaling (hsa04350; *p*_adj_ < 0.05), which was found to be enriched at 18% O_2_. In Huh-7 cells, pathways such as extracellular matrix (ECM) organization (R-HSA-1474244; *p*_adj_ < 0.005) and drug metabolism by the cytochrome P450 (CYP450) enzymes (hsa00982; *p*_adj_ < 0.05) were strongly enriched at 5% O_2_, while oxidative phosphorylation (hsa00190; *p*_adj_ < 0.005) and oxidative stress-induced senescence (R-HSA-2559580; *p*_adj_ < 0.05) were enriched at 18% O_2_. Interestingly, in contrast to Huh-7 cells, both PC-3 and SH-SY5Y showed enrichment of annotation terms related to mitochondrial respiration and oxidative phosphorylation at 5% O_2_ (see Figure 2). Signaling by interleukins (R-HSA-449147; *p*_adj_ < 0.05) and neurogenesis (GO:0022008; *p*_adj_ < 0.005) were among the processes enriched at 18% O_2_ in SH-SY5Y cells. The full lists of functional annotation terms enriched by O_2_ level in all cell lines are available in Appendix A.

### 3.2. The Top Differentially Expressed Genes Have Key Roles in Cancer Cell Biology

By sorting the DEGs according to their adjusted *p*-value, we found that most of the genes highly affected by O_2_ are implicated in cancer cell biology, including several with roles in cancer cell proliferation, tumor progression, metastasis, invasion, and chemosensitivity to anticancer therapy. A selection of these genes is shown in Table 1. The complete list of top 10 protein-coding DEGs in all cell lines at both O_2_ conditions and their corresponding log_2_FC values are shown in Figure 3.

### 3.3. Oxygen Levels Affected mtDNA-Encoded Transcript Abundances in Most Cell Lines

Oxygen induced differential expression of mtDNA-encoded genes in most cell lines (Table 2). In Huh-7 cells, 11 mtDNA-encoded gene transcripts were affected by O_2_, of which six are subunits of respiratory complexes I, IV, and V, while the rest are mitochondrial tRNAs. Interestingly, all of these were upregulated at 18% O_2_. Eleven mtDNA-encoded genes were affected by O_2_ in PC-3 cells and 10 in SH-SY5Y, however, all were upregulated at 5% O_2_, in striking contrast with the observation in Huh-7 cells. Again, these DEGs encoded subunits of the respiratory chain and tRNAs. In contrast, only two mtDNA-encoded genes were affected in LNCaP. These results suggest that O_2_ levels in cell culture affect the expression of mtDNA-encoded genes, but in a highly cell-type specific manner.

### 3.4. HIF Targets Were Upregulated at 5% O_2_ in All Cell Lines

Five percent O_2_ is not hypoxic, and we have previously shown that, under the conditions used here, pericellular O_2_ levels do not fall below 3.7%, even during extensive static incubation periods in culture [10]. A handful of reports have shown that HIF-1/2 activity is detectable at 2–5% O_2_ levels (i.e., physioxia) [6,29,30]. Here we identified several HIF-1/2 gene targets upregulated at 5% O_2_ in all cell lines, a selection of which is shown in Table 3. For example, in LNCaP, transcripts related to angiogenesis and vasodilation, such as *VEGFA* and *ADM*, were enriched. Similarly, genes that encode enzymes involved in the metabolic reprograming of cells towards a glycolytic phenotype were upregulated in Huh-7 cells grown at 5% O_2_. These genes include the glucose transporter *SLC2A3* (GLUT3), the glycolytic enzyme *ENO2* (enolase), and the gluconeogenic enzyme *PCK1* (phosphoenolpyruvate carboxykinase 1). A lower number of HIF-1/2 targets were detected in PC-3 and SH-SY5Y cells, consistent with our initial observation that these two cell lines were less sensitive to O_2_ than Huh-7 and LNCaP cells.

We next investigated whether the HIF targets upregulated at 5% O_2_ in our cells were regulated by HIF-1α, HIF-2α, or both. To do this, we compared our DEG datasets with the dataset from the study by Downes et al. who sequenced the transcriptional outputs of stabilized forms of HIF-1α and HIF-2α [75]. We obtained a more exhaustive list of HIF-1/2 targets upregulated at 5% O_2_ in our cell lines (Appendix A). A total of 103, 145, 40, and 29 HIF-1/2 targets were identified in LNCaP, Huh-7, PC-3, and SH-SY5Y, respectively. When comparing the proportion of unique and shared DEGs regulated by HIF-1α and HIF-2α, we found that the number of unique HIF-2α targets was greater in all cell lines, although several gene targets of both were also observed (Figure 4).

### 3.5. Expression Patterns of Genes Upregulated at 5% O_2_ in LNCaP Cells, but Not PC-3, Better Correlate with the In Vivo Prostate Cancer Transcriptome

To determine if physiological O_2_ levels in cell culture produce a transcriptional signature that better resembles a prostate cancer transcriptome in vivo, we compared our transcriptomic data from LNCaP and PC-3 cells (two prostate adenocarcinoma cell lines) with the expression data from the Prostate Cancer Transcriptome Atlas (PCTA), which contains a dataset of expression levels of 18,390 genes from 2,115 human prostate tumor samples [19]. We independently matched the DEGs upregulated at 5% O_2_ and at 18% O_2_ from both cell lines with the PCTA dataset. Upon obtaining matched gene lists from our data and the PCTA dataset, we performed a correlation analysis of the change in gene expression levels (log_2_FC) in both datasets. In LNCaP cells, gene expression patterns from the DEGs upregulated at 5% O_2_ showed a highly statistically significant (*p* < 0.0001) albeit weak (r = 0.1837) correlation with the PCTA expression data (Figure 5A). On the other hand, expression levels of DEGs upregulated at 18% O_2_ showed no statistically significant correlation (*p* = 0.1331) with the PCTA expression data (Figure 5B). Finally, the correlation between the expression levels of DEGs upregulated at either 5% O_2_ or 18% O_2_ in PC-3 cells, and the PCTA dataset was non-significant (*p* > 0.05).

## 4. Discussion

The main goal of this study was to determine how O_2_ tension in the standard cell culture environment (18% O_2_) impacts the cancer cell transcriptome compared to a more representative in vivo environment (5% O_2_). We were particularly interested in the extent to which any effects of O_2_ were shared amongst cell lines versus cell-line specific. Our results indicate broad transcriptional effects of O_2_ between 5% and 18% that are highly cell-type specific. Even in LNCaP and PC-3, both prostate cancer cell lines, the overlap in O_2_-dependent DEGs was only ~5%. These results are consistent with a previous study showing little overlap in the proteome of three diffuse large B-cell lymphoma cell lines cultured in the same two O_2_ conditions [76]. Increased oxidative stress associated with higher O_2_ levels affects a variety of pathways, including the p53 pathway and mitogen-activated protein kinase (MAPK) pathways (reviewed in [77]). Given their different origins and genetic backgrounds, cancer cells have distinct mutations, including gene copy number differences, of genes related to these pathways. Therefore, differential transcriptional response of these cell lines to O_2_ is perhaps not surprising. In any case, these observations highlight the need for considering oxygenation status as an important factor in experimental design, since the effects of growing cells at 18% O_2_ are broad and may not be easy to predict.

Functional enrichment analysis revealed that biological processes and pathways relevant to the disease etiology of each cell type were altered by O_2_ level. For example, prostate cancer commonly metastasizes to bone, forming primarily osteoblastic lesions. TGF-β and bone morphogenetic proteins (BMPs), released by prostate cancer cells, induce osteoblast differentiation, which in turn releases growth factors that stimulate the proliferation of prostate cancer cells [78]. The TGF-β signaling pathway (hsa04350) was the annotation term most significantly enriched (*p*_adj_ < 0.05) in LNCaP cells at 18% O_2_. Signaling by BMP (R-HSA-201451) and osteoblast differentiation (GO:0001649) were also enriched (*p* < 0.05). Another interesting observation was the upregulation of the androgen receptor (*AR* gene) in PC-3 cells grown at 18% O_2_ (see Appendix A), which is contrasting with fact that PC-3 is derived from an androgen-insensitive tumor and do not express *AR*. These interactions of key prostate cancer genes and signaling pathways with O_2_ levels attest to the potential issues associated with a hyperoxic cell culture environment. Functional studies are necessary to determine how O_2_ levels in cell culture affect metastasis and other functional characteristics of prostate cancer cells in vitro

One of the main functions of hepatic cells is detoxification of xenobiotics through phase I (CYP450) and phase II enzymes. In Huh-7 cells, drug metabolism by the CYP450 (hsa00982) was among the annotation terms enriched at 5% O_2_ (*p*_adj_ < 0.05), in accordance with a previous observation that *CYP1A1*, *CYP1A2*, and *CYP2E1*, along with a number of phase II enzymes, were upregulated in HepG2 cells cultured in physioxia, compared with cells at atmospheric O_2_ [79]. Differential expression of phase I and II enzymes at different O_2_ tensions leads to altered hepatic metabolism of drugs and toxins, which in turn results in altered biological responses to these compounds when tested in vitro. Indeed, DiProspero et al. showed that the toxicity and potency of acetaminophen, cyclophosphamide, and aflatoxin B1 were dependent on O_2_ tension in HepG2 cells cultured at ~18% O_2_, 8% O_2_, and 3% O_2_ [79]. Notably, these pharmacodynamic parameters obtained from cells grown at physioxic conditions better matched in vivo primary human hepatocyte data than cells cultured under standard conditions. Therefore, O_2_ tension should be considered as an important factor in the design of experiments aimed to study the effects and mechanisms of bioactive molecules in vitro.

We observed several HIF-1/2 targets upregulated at 5% O_2_ in LNCaP and Huh-7 cells. Although traditionally known as transcription factors that mediate the response to hypoxia, some studies have shown HIF-1/2 expression and activity in the physiological O_2_ range [6,29,30]. A thorough discussion of the roles of HIF and other pathways in physioxia versus normoxia (18% O_2_) is provided in [8]. The fact that fewer HIF-1/2 targets upregulated at 5% O_2_ were observed in PC-3 and SH-SY5Y cells supports the notion that these cells are less sensitive to O_2_ than LNCaP and Huh-7. Moreover, because cells were grown at either 5% O_2_ or 18% O_2_ for 14 days, the induction of HIF targets shown in this study suggests a role for HIF activity in physioxia, rather that it being a mere consequence of an acute reduction in O_2_ levels while performing the experiments. Similar observations have been made in other studies [30,80], supporting this notion. Remarkably, the proportion of unique HIF-2α targets, compared to HIF-1α targets, was higher in all the cell lines (see Figure 4). These results are in line with previous observations where HIF-1α is described as the most active isoform in acute and severe hypoxia, whereas HIF-2α has been shown to be predominantly active during chronic and “physiological” hypoxia [81,82]. Downes et al. showed through functional enrichment analysis that the genes induced by HIF-1α are associated with biological processes like glycolysis and NADH regeneration, while HIF-2α–enriched processes include angiogenesis, extracellular matrix organization, pattern specification process, and negative regulation of cell adhesion. Indeed, GO terms and KEGG pathways related to all the above processes were found to be enriched by the HIF–regulated genes upregulated at 5% O_2_ in our cell lines (see Appendix A). In addition, given the variety of mechanisms in tumorigenesis regulated by HIF (e.g., metabolism, migration, invasion, survival), complete loss of HIF-1/2 activity at 18% O_2_ may compromise experiments focused on cancer biology and chemotherapeutic strategies.

Enrichment of functional annotation terms related to the mitochondria and/or mitochondrial processes (e.g., respiration) was observed in three of the four cell lines (Huh-7, PC-3 and SH-SY5Y), although the directionality of the effects were inconsistent. While mitochondrial terms were enriched at 18% O_2_ in Huh-7 cells, they were enriched at 5% O_2_ in PC-3 and SH-SY5Y. The same general trend was apparent for mtDNA–encoded genes. Decreased expression of mtDNA–encoded genes in Huh-7 cells at 5% O_2_ may reflect decreased mitochondrial abundance. Indeed, mitochondrial biogenesis (R-HSA-1592230) was one of the Reactome pathways enriched at 18% O_2_ in Huh-7 cells (*p*_adj_ < 0.05; see Appendix A). In agreement, Moradi et al. observed decreased mitochondrial footprint in Huh-7 cells grown at 5% O_2_ compared to 18% O_2_ in the same conditions [5]. Further, it has been shown that HEY1, a HIF-1 target, decreases mitochondrial biogenesis by repressing the expression of PTEN-induced kinase 1 (PINK1) in human hepatocellular carcinoma cells through a HIF-1–dependent mechanism. In accordance, our data shows *HEY1* expression increased 2.05-fold at 5% O_2_ (*p* < 0.05; see Table 3). Additionally, Zhang et al. reported that HIF-1 inhibits mitochondrial biogenesis in renal carcinoma cells by repressing PGC-1β [83]. Interestingly, expression of *PPARGC1B* was ~3 times lower in Huh-7 cells at 5% O_2_ compared to 18% (*p*_adj_ < 0.005; see Appendix A). On the other hand, expression of all mtDNA–encoded genes affected by O_2_ was higher in physioxia in both PC-3 and SH-SY5Y cells. Expression of genes related to mitochondrial biogenesis induced by HIF-1 has been observed in the neuroblastoma cell line SK-N-AS when exposed to hypoxia, along with increased mtDNA copy numbers [84]. Moreover, mitochondrial abundance was found to be higher in primary neurons grown at 2% and 5% O_2_ compared to atmospheric O_2_ [85]. No direct link between O_2_ tension and regulation of mitochondrial biogenesis has been reported in prostate cancer. Future research should be directed to investigating the O_2_–dependent mechanisms of mitochondrial gene expression and biogenesis in different cell types.

Another aspect of cancer biology that has been shown to be affected by O_2_ levels is cancer cell stemness [86]. As happens with non-transformed stem cells, the stemness of cancer cells is regulated by transcription factors like the octamer-binding transcription factor 4 (Oct4), Nanog, and SRY-box 2 (Sox2) [87]. Crosstalk between the HIF-1/2 pathway and the Oct4/Nanog/Sox2 axis is well documented [86]. For instance, Li et al. demonstrated that HIF-1/2 regulates the tumorigenic capacity of glioma stem cells and showed that HIF-2α colocalizes with cancer stem cell markers [88]. Using data from the study by Sharov et al. [89], we analyzed the effects of O_2_ levels in our experimental groups on the expression of Oct4/Nanog/Sox2 related genes. While the expression of Oct4/Nanog/Sox2 was not affected by O_2_ levels in most of the cell lines used here—except SH-SY5Y where *POU5F1* (Oct4) was upregulated at 5% O_2_—we did observe differential expression of their gene targets at 5% and 18% O_2_ (see Appendix A). A similar observation was made by Westfall et al. where they observed constant expression of Oct4/Nanog/Sox2, but differential expression of some of their targets in human embryonic stem cells grown in 4% O_2_ versus 18% O_2_ [90]. Altogether, these results suggest that culturing cancer cells at non-physiological O_2_ levels may alter the transcriptional programs controlling stemness and renewal, and may in turn affect the response of cancer cells to chemotherapeutic drugs in vitro.

Finally, we analyzed the correlation between the transcriptomic data from the two prostate cancer cell lines used here and in vivo prostate tumor transcriptional profiles, by using the dataset from the PCTA [19,91]. In LNCaP cells, we found that the list of DEGs upregulated in physioxia (5% O_2_) shows a highly significant correlation (*p* < 0.0001) with the PCTA gene expression data, whereas the DEGs upregulated at 18% O_2_ has no significant correlation with the PCTA data. The overall weak correlation degree (Pearson r = 0.1837) of DEGs upregulated in physioxia and PCTA data could be explained by the fact that LNCaP is derived from a metastatic tumor from a single patient whereas the PCTA is a collection of data from a total of 2,115 patients. This shows the high heterogeneity of cancers within different patients. No significant correlation with the PCTA data was found from the transcriptomic profiles of PC-3 cells at either O_2_ level, which supports the notion that PC-3 cells are not as sensitive to O_2_ tension (between 5–18% O_2_) as LNCaP and Huh-7 cells. Additional research should be conducted using multi-omics approaches to investigate how culturing cells in physioxia produces more similar genomic, transcriptomic, proteomic, and epigenomic profiles to the ones observed in tissues in vivo.

This study has some limitations. We recognize that transcriptional effects do not necessarily reflect changes in the cell proteome or phenotype, and this may be due to complex epigenetic, post-transcriptional, and post-translational mechanisms governing gene expression and protein function [92,93]. Investigation of the effects of O_2_ levels on the epigenome, proteome, metabolome, and lipidome of different cell types is certainly warranted. In addition, data obtained from functional enrichment analysis needs to be interpreted carefully, as their significance in terms of cell behavior is not always clear. As an example, “regulation of apoptotic process” (GO:0042981) is one of the biological processes (GO) found to be enriched in PC-3 cells at 18% O_2_, with 11 DEGs associated with this process being upregulated at this O_2_ tension. Out of these eleven genes, five have an antiapoptotic role, five are proapoptotic, and one has dual roles in the regulation of apoptosis. Thus, the functional effect of 18% O_2_ in the regulation of apoptosis in PC-3 cells on apoptosis cannot be predicted by our bioinformatic analysis alone, and an experiment challenging PC-3 cells to apoptotic stimuli at both 5% O_2_ and 18% O_2_ would be necessary. Nonetheless, our results provide insight into the cellular processes and pathways that may be affected by O_2_ in these cancer cells, which can direct subsequent research questions.

In conclusion, our results show that supraphysiological O_2_ levels in cell culture significantly alter the global transcriptomes of cancer cell lines in highly cell-line specific ways. This makes it difficult to establish general rules regarding how non-physiological O_2_ levels might affect experiments. Our results, together with the increasing amount of functional data regarding the effects of physioxia versus standard cell culture hyperoxia [8], should encourage cell culturists to implement the regulation of O_2_ levels in their experiments. This would certainly be expected to increase the likelihood that results will translate to in vivo.

## Figures and Tables

**Figure 1 biomolecules-12-01684-f001:**
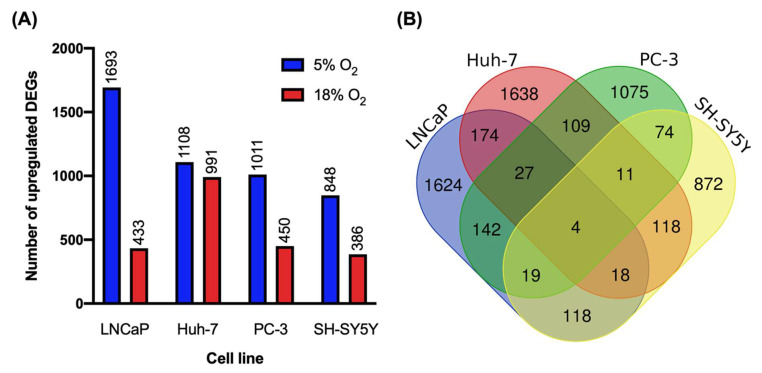
**(A**) Number of differentially expressed genes (DEGs) upregulated at 5% and 18% O_2_ in each cell line. (**B**) Venn diagram showing the overlap of all DEGs affected by O_2_ among the cell lines.

**Figure 2 biomolecules-12-01684-f002:**
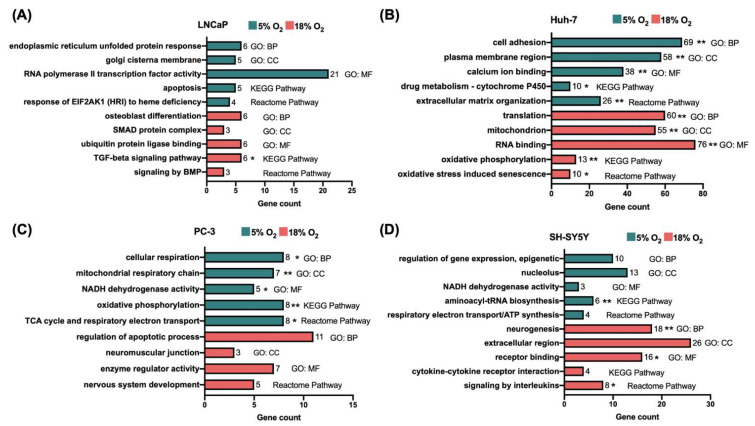
Selection of functional annotation terms enriched at 5% and 18% O_2_ in (**A**) LNCaP, (**B**) Huh-7, (**C**) PC-3, and (**D**) SH-SY5Y cells. GO, gene ontology; CC, cellular component; MF, molecular function; KEGG, Kyoto encyclopedia of genes and genomes. (* *p*_adj_ < 0.05, ** *p*_adj_ < 0.005, otherwise *p* < 0.05). Other abbreviations: BMP, bone morphogenetic protein; EIF2AK1, eukaryotic translation initiation factor 2 alpha kinase 1; HRI, heme-regulated inhibitor; SMAD, SMA (small)—mothers against decapentaplegic; TCA, tricarboxylic acid; TGF, transforming growth factor.

**Figure 3 biomolecules-12-01684-f003:**
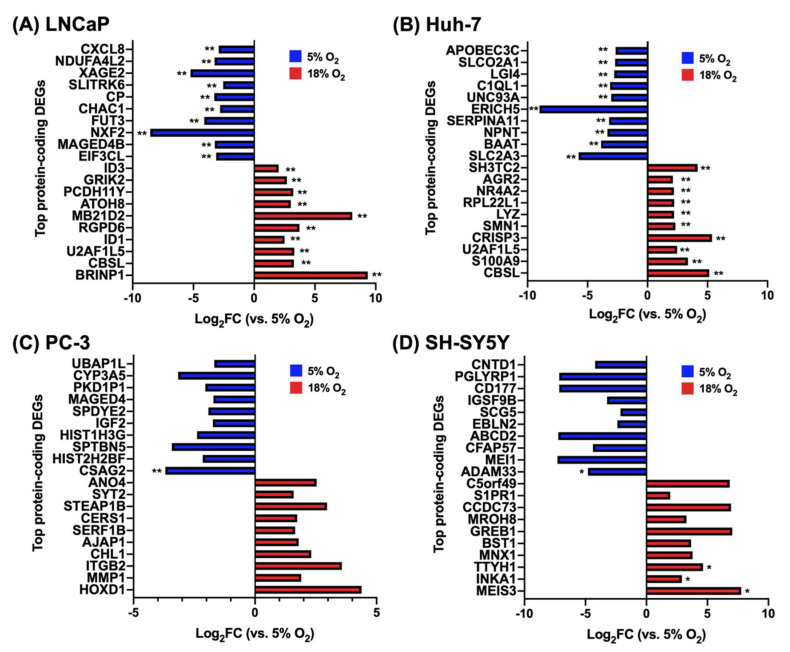
Top 10 protein-coding differentially expressed genes (DEGs) in (**A**) LNCaP cells, (**B**) Huh-7 cells, (**C**) PC-3 cells, and (**D**) SH-SY5Y cells at 18% O_2_ vs 5% O_2_. (* *p*_adj_ < 0.05, ** *p*_adj_ < 0.005, otherwise *p* < 0.05).

**Figure 4 biomolecules-12-01684-f004:**
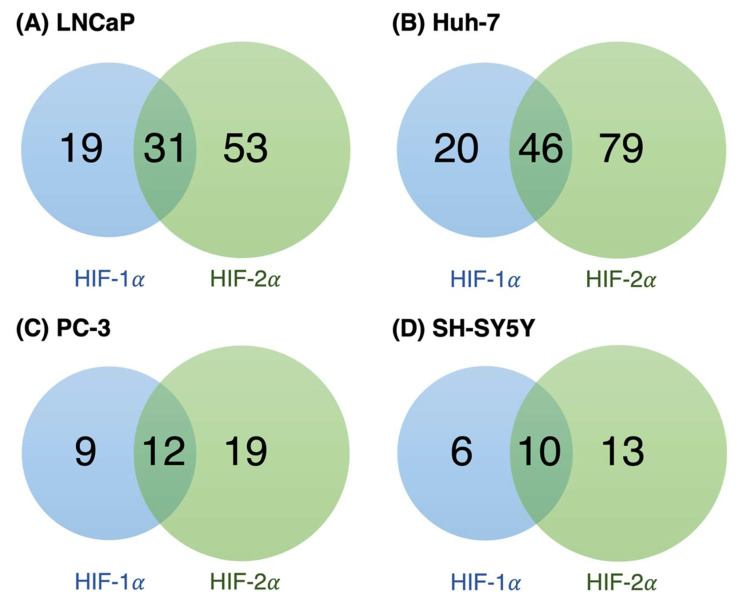
Venn diagrams showing the proportion of unique and shared HIF-1α and HIF-2α gene targets upregulated at 5% O_2_ in (**A**) LNCaP, (**B**) Huh-7, (**C**) PC-3, and (**D**) SH-SY5Y cells. Hypoxia-inducible targets and their regulation by HIF-1/2 were identified by matching our data with the dataset from [75].

**Figure 5 biomolecules-12-01684-f005:**
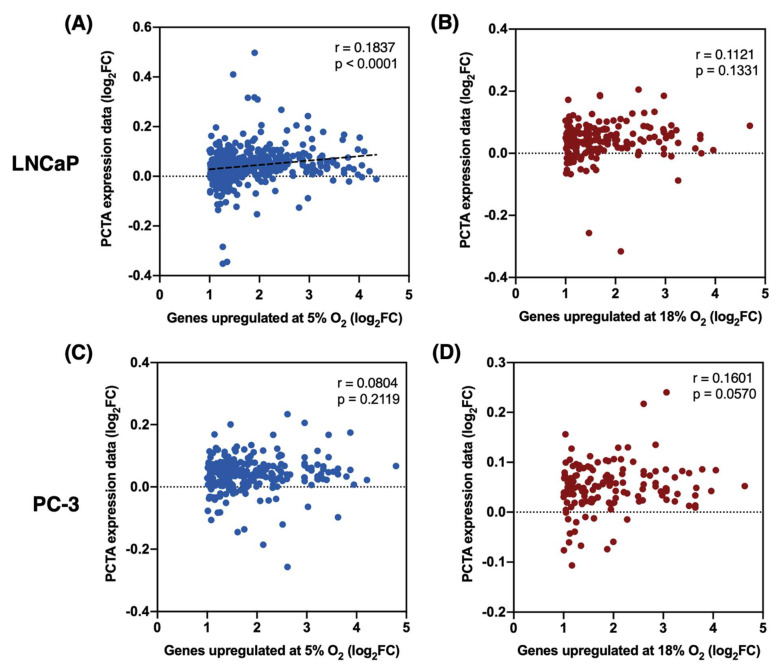
Correlation between gene expression levels (log_2_FC) between PCTA expression data and DEGs upregulated at (**A**) 5% O_2_ and (**B**) 18% O_2_ in LNCaP cells, and DEGs upregulated at (**C**) 5% O_2_ and (**D**) 18% O_2_ in PC-3 cells.

**Table 1 biomolecules-12-01684-t001:** Selected DEGs at 18% O_2_ vs 5% O_2_ with important roles in cancer biology.

Gene Symbol	Gene Name	Role in Cancer Biology	Log_2_FC ^†^	Refs.
**LNCaP**				
*ID1, ID3* **	Inhibitor of DNA binding 1 and Inhibitor of DNA binding 3	Transcription factor repressors; mediate metastasis, androgen resistance, and chemoresistance.	+2.50, +2.02	[21]
*CHAC1* **	ChaC glutathione specific gamma-glutamylcyclotransferase 1	Degrades glutathione. Involved in ferroptosis; associated with increased chemosensitivity.	−2.81	[22]
**Huh-7**				
*S100A9* **	S100 calcium binding protein A9	TLR4 and RAGE ligand, promotes HCC progression through MAPK and NF-κB pathways.	+3.37	[23]
*SLC3A2* ** (GLUT3)	Solute carrier family 2 member 3	Selective glucose uniporter. Expression is correlated with HCC growth/invasion.	−5.74	[24]
**PC-3**				
*GREB1*	Growth regulating estrogen receptor binding 1	Regulated by androgens, contributes to prostate cancer growth and antiandrogen resistance.	+7.03	[25]
*ADAM33* *	ADAM metallopeptidase domain 33	Methylation and upregulation observed in breast cancer.	−4.79	[26]
**SH-SY5Y**				
*MMP1*	Matrix metallopeptidase 1	Upregulated in a wide variety of cancer types.	+1.90	[27]
*CSAG2* ** (TRAG-3)	CSAG family member 2	First isolated from taxol-resistant ovarian cancer cell line. Overexpressed in many cancer types, correlated with tumor progression.	−3.69	[28]

* *p*_adj_ < 0.05, ** *p*_adj_ < 0.005, otherwise *p* < 0.05. ^†^ Positive value means gene is upregulated at 18% O_2_ while negative value indicates upregulation at 5% O_2_. Abbreviations: GLUT3, glucose transporter 3; HCC, hepatocellular carcinoma; MAPK, mitogen-activated protein kinase; NF-κB, nuclear factor kappa B; RAGE, receptor for advanced glycation end-products; TLR4, toll-like receptor 4; TRAG-3, taxol-resistance associated gene 3.

**Table 2 biomolecules-12-01684-t002:** Mitochondrially-encoded genes differentially expressed at 18% versus 5% O_2_.

Gene Symbol	Gene Name	Description/Role	Log_2_FC ^†^
**LNCaP**			
*MT-TL2*	Mitochondrially encoded tRNA leucine 2 (CUN)	Transfer RNA for leucine	−1.21
*MT-TW*	Mitochondrially encoded tRNA tryptophan	Transfer RNA for tryptophan	+1.82
**Huh-7**			
*MT-ND1*	Mitochondrially encoded NADH:ubiquinone oxidoreductase core subunit 1	Complex I subunit	+1.36
*MT-CO1*	Mitochondrially encoded cytochrome c oxidase I	Complex IV subunit	+1.07
*MT-CO2*	Mitochondrially encoded cytochrome c oxidase II	Complex IV subunit	+1.19
*MT-CO3*	Mitochondrially encoded cytochrome c oxidase III	Complex IV subunit	+1.13
*MT-ATP6*	Mitochondrially encoded ATP synthase membrane subunit 6	ATP synthase subunit	+1.07
*MT-ATP8*	Mitochondrially encoded ATP synthase membrane subunit 8	ATP synthase subunit	+1.03
*MT-TY **	Mitochondrially encoded tRNA tyrosine	Transfer RNA for tyrosine	+1.94
*MT-TL1*	Mitochondrially encoded tRNA leucine 1 (UUA/G)	Transfer RNA for leucine	+1.44
*MT-TV*	Mitochondrially encoded tRNA valine	Transfer RNA for valine	+2.07
*MT-TW*	Mitochondrially encoded tRNA tryptophan	Transfer RNA for tryptophan	+3.14
*MT-TT*	Mitochondrially encoded tRNA threonine	Transfer RNA for threonine	+1.00
**PC-3**			
*MT-ND2*	Mitochondrially encoded NADH:ubiquinone oxidoreductase core subunit 2	Complex I subunit	−1.08
*MT-ND4*	Mitochondrially encoded NADH:ubiquinone oxidoreductase core subunit 4	Complex I subunit	−1.04
*MT-ND4L*	Mitochondrially encoded NADH:ubiquinone oxidoreductase core subunit 4L	Complex I subunit	−1.14
*MT-ND5*	Mitochondrially encoded NADH:ubiquinone oxidoreductase core subunit 5	Complex I subunit	−1.01
*MT-ND6*	Mitochondrially encoded NADH:ubiquinone oxidoreductase core subunit 6	Complex I subunit	−1.04
*MT-CYB*	Mitochondrially encoded cytochrome b	Complex III subunit	−1.07
*MT-CO2*	Mitochondrially encoded cytochrome c oxidase II	Complex IV subunit	−1.01
*MT-ATP8*	Mitochondrially encoded ATP synthase membrane subunit 8	ATP synthase subunit	−1.18
*MT-TA*	Mitochondrially encoded tRNA alanine	Transfer RNA for alanine	−3.33
*MT-TL1*	Mitochondrially encoded tRNA leucine 1 (UUA/G)	Transfer RNA for leucine	−1.07
*MT-TM*	Mitochondrially encoded tRNA methionine	Transfer RNA for methionine	−2.29
**SH-SY5Y**			
*MT-ND3*	Mitochondrially encoded NADH:ubiquinone oxidoreductase core subunit 3	Complex I subunit	−1.13
*MT-ND5*	Mitochondrially encoded NADH:ubiquinone oxidoreductase core subunit 5	Complex I subunit	−1.02
*MT-ND6*	Mitochondrially encoded NADH:ubiquinone oxidoreductase core subunit 6	Complex I subunit	−1.01
*MT-ATP8*	Mitochondrially encoded ATP synthase membrane subunit 8	ATP synthase subunit	−1.07
*MT-TH*	Mitochondrially encoded tRNA histidine	Transfer RNA for histidine	−2.01
*MT-TE*	Mitochondrially encoded tRNA glutamic acid	Transfer RNA for glutamate	−1.99
*MT-TG*	Mitochondrially encoded tRNA glycine	Transfer RNA for glycine	−1.31
*MT-TQ*	Mitochondrially encoded tRNA glutamine	Transfer RNA for glutamine	−6.27
*MT-TT*	Mitochondrially encoded tRNA threonine	Transfer RNA for threonine	−1.38
*MT-TS2*	Mitochondrially encoded tRNA serine 2 (AGU/C)	Transfer RNA for serine	−1.27

* *p*_adj_ < 0.05, otherwise *p* < 0.05. ^†^ Positive value means gene is upregulated at 18% O_2_ while negative value indicates upregulation at 5% O_2_.

**Table 3 biomolecules-12-01684-t003:** Selection of differentially expressed HIF-1/2 targets upregulated at 5% O_2_.

Gene Symbol	Gene Name	Role	Log2FC	Refs. ^†^
**LNCaP**				
*VEGFA* *	Vascular endothelial growth factor A	Promotes angiogenesis	1.68	[31]
*ADM*	Adrenomedullin	Vasodilator peptide	1.62	[32]
*CALCRL*	Calcitonin receptor like receptor	G protein-coupled receptor related to the calcitonin receptor; enables adrenomedullin binding activity	1.63	[33]
*ADORA2A* *	Adenosine A2a receptor	Activates adenylyl cyclase, inducing cAMP signaling	2.19	[34]
*NDUFA4L2* **	NDUFA4, mitochondrial complex associated like 2	Complex I subunit; shown to decrease O_2_ consumption	3.26	[35]
*PLOD2* **	Procollagen-lysine,2-oxoglutarate 5-dioxygenase 2	Catalyzes the hydroxylation of lysyl residues in collagen-like peptides	7.79	[36]
*LOX*	Lysyl oxidase	Facilitates the crosslinking of collagens and elastin	1.60	[37]
*CP* **	Ceruloplasmin	Involved in Cu transport	3.28	[38]
*TF*	Transferrin	Involved in Fe transport	6.37	[39]
*PMAIP1*	Phorbol-12-myristate-13-acetate-induced protein 1	Pro-apoptotic protein	1.18	[40]
*ENG*	Endoglin	Auxiliary receptor for the TGF-β receptor complex	1.43	[41]
*STC2*	Stanniocalcin 2	May have autocrine and paracrine functions; may be involved in Ca^2+^ and phosphate transport and metabolism	1.19	[42]
*GPX8*	Glutathione peroxidase 8	Catalyzes reduction of hydrogen and alkyl peroxides	1.19	[43]
*CXCL12*	C-X-C motif chemokine ligand 12	Chemoattractant cytokine	1.67	[44]
**Huh-7**				
*SLC2A3* **	Solute carrier family 2 member 3 (GLUT3)	Selectively transports glucose into the cytosol	5.74	[45]
*SLC2A14 ***	Solute carrier family 2 member 14 (GLUT14)	Transports glucose into the cytosol	2.64	[46]
*HKDC1*	Hexokinase domain containing 1	Novel member of the hexokinase family, involved in glucose metabolism	1.51	[47]
*PCK1*	Phosphoenolpyruvate carboxykinase 1	Catalyzes conversion of PEP to oxaloacetate during gluconeogenesis	1.49	[48]
*ENO2*	Enolase 2	Catalyzes conversion of 2-phosphoglycerate to PEP during glycolysis	1.38	[49]
*COX4I2*	Cytochrome c oxidase subunit 4I2	Complex IV subunit; regulates efficiency of electron transport and O_2_ consumption	6.04	[50]
*CP* **	Ceruloplasmin	Involved in Cu transport	1.84	[38]
*LOXL2* *	Lysyl oxidase like 2	Facilitates the crosslinking of collagens and elastin	1.66	[37]
*LOXL4* **	Lysyl oxidase like 4	Facilitates the crosslinking of collagens and elastin	1.93	[51]
*P4HA2*	Prolyl 4-hydroxylase subunit alpha 2	Catalyzes the formation of 4-hydroxyproline during collagen synthesis	1.14	[36]
*PLOD2*	Procollagen-lysine,2-oxoglutarate 5-dioxygenase 2	Catalyzes the hydroxylation of lysyl residues in collagen-like peptides	1.08	[36]
*NPPB* **	Natriuretic peptide B	Hormone that mediates natriuresis, diuresis, and vasodilation	2.05	[52]
*EPO* *	Erythropoietin	Promotes erythropoiesis	1.45	[53]
*PDGFB* **	Platelet derived growth factor subunit B	Potent mitogen and chemoattractant, promotes angiogenesis	2.06	[54]
*CXCL6*	C-X-C motif chemokine ligand 6	Chemoattractant cytokine	1.21	[55]
*IGFBP1* **	Insulin like growth factor binding protein 1	Binds insulin-like growth factors, promotes migration and metabolism	2.66	[56]
*TXNIP ***	Thioredoxin interacting protein	Binds to and inhibits thioredoxin	2.73	[57]
*NDRG1* *	N-myc downstream regulated 1	Involved in p53-mediated caspase activation and apoptosis	1.53	[58]
*PTPRR **	Protein tyrosine phosphatase receptor type R	Membrane receptor, regulates cell cycle, differentiation and oncogenesis	1.94	[59]
*NR4A3 **	Nuclear receptor subfamily 4 group A member 3	Nuclear receptor and transcriptional activator	6.86	[60]
*EGLN3*	egl-9 family hypoxia inducible factor 3 (PHD-3)	Catalyzes hydroxylation of HIFs for subsequent degradation	2.27	[61]
*TFF2*	Trefoil factor 2	May stabilize the mucus layer and affect healing of the epithelium	1.19	[62]
*HEY1*	Hes related family bHLH transcription factor with YRPW motif 1	Transcriptional repressor; inhibits mitochondrial biogenesis in HCC	1.06	[63]
**PC-3**				
*SLC2A9*	Solute carrier family 2 member 9 (GLUT9)	Transports glucose into the cytosol	5.73	[64]
*PDK1*	Pyruvate dehydrogenase kinase 1	Phosphorylates and inhibits the pyruvate dehydrogenase complex	1.03	[65]
*CA9*	Carbonic anhydrase 9	Catalyzes interconversion between CO_2_ and H_2_O into carbonic acid	1.98	[66]
*TERT*	Telomerase reverse transcriptase	Mediates extension and replenishment of telomeres	1.98	[67]
*TH*	Tyrosine hydroxylase	Catalyzes the conversion of tyrosine to dopamine	1.09	[68]
*BNIP3*	BCL2 interacting protein 3	Pro-apoptotic factor	1.02	[69]
*IL33*	Interleukin 33	Proinflammatory cytokine	1.38	[70]
*PPP1R3C*	Protein phosphatase 1 regulatory subunit 3C	Subunit of the protein phosphatase 1 complex; modulates glycogen metabolism	6.37	[71]
**SH-SY5Y**				
*IGF2*	Insulin like growth factor 2	Promotes growth and proliferation	1.74	[72]
*PPP1R3C*	Protein phosphatase 1 regulatory subunit 3C	Subunit of the protein phosphatase 1 complex; modulates glycogen metabolism	1.01	[71]
*ABCB6*	ATP binding cassette subfamily B member 6	ABC transporter; plays a role in porphyrin transport	1.25	[73]
*TRIM29*	Tripartite motif containing 29	Transcriptional regulatory factor involved in carcinogenesis and/or differentiation	2.78	[74]

* *p*_adj_ < 0.05, ** *p*_adj_ < 0.005, otherwise *p* < 0.05. Abbreviations: ABC, ATP-binding cassette; cAMP, cyclic adenosine monophosphate; BCL2, B-cell lymphoma 2; GLUT, glucose transporter; HCC, hepatocellular carcinoma; HIF; hypoxia-inducible factor; OMM, outer mitochondrial membrane; PEP, phosphoenolpyruvate; PHD-3, prolyl hydroxylase 3; ROS, reactive oxygen species; TGF-β, transforming growth factor-beta. ^†^ References identifying the gene as a HIF-1/2 target.

## Data Availability

The raw RNA-seq data has been deposited to the NCBI SRA database (BioProject PRJNA871952). SRA accession numbers: SRR21155729, SRR21155728, SRR21155727, SRR21155733, SRR21155732, SRR21155731, SRR21155730, and SRR21155726.

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
