# Peer review of "Culture of Cancer Cells at Physiological Oxygen Levels Affects Gene Expression in a Cell-Type Specific Manner"

_biomolecules, 2022, doi:10.3390/biom12111684_

Round 1

Reviewer 1 Report

Overall this was a thorough and interesting study into the effect of oxygen tension on 4 cancer cell lines. It opens the door for many interesting studies and promotes the use of a more physiological oxygen tension in cell culture. There are several areas which should be addressed, the most crucial being point 7.

11)      Not clear why you selected two prostate cancer lines, a neuroblastoma and hepatocarcinoma for study

22)      How did you arrive at 5% oxygen tension being physioxic for these cancers? 5% is physioxic for cartilage - https://www.frontiersin.org/articles/10.3389/fbioe.2020.590743/full but cartilage lacks blood vessels. A 2% level was found to be beneficial for adipose derived cells https://stemcellres.biomedcentral.com/articles/10.1186/s13287-018-0891-4 Even brief periods of atmospheric oxygen changed tumor biology vs. 3% oxygen tension https://www.science.org/doi/10.1126/sciadv.abh3375

33)      Was subculture etc. performed in a glove box (i.e. controlled oxygen environment) or in atmospheric conditions?

44)      What cell passage was used in each case?

55)      Line 84 – was overall transcript abundance the same, increased or decreased by oxygen tension?

66)      Fig 2 – some terms are lower case, others upper, please be consistent. Also, resolution of the words is low.

77)      Figure and table legends indicate that the adjusted p value was not used as a cut off for inclusion. If the adjusted p value is not less than 0.05 it does not seem appropriate to include that data. This artificially inflates the number of differentially expressed genes.

Author Response

Response to the 1st round of review comments

Reviewer 1

Comments and Suggestions for Authors

Overall this was a thorough and interesting study into the effect of oxygen tension on 4 cancer cell lines. It opens the door for many interesting studies and promotes the use of a more physiological oxygen tension in cell culture. There are several areas which should be addressed, the most crucial being point 7.

1)      Not clear why you selected two prostate cancer lines, a neuroblastoma and hepatocarcinoma for study.
Response: The questions we asked in this study were general and could have been addressed with any cell lines: firstly, how widespread are the effects of 18% O2, relative to physioxia, in cell culture? Secondly, are these effects common to all cell lines, or highly specific to an individual cell line? But since we have previously investigated the effects of 18% versus 5% O2 on specific cellular activities, like energy metabolism, mitochondrial network characteristics1 and response to drugs such as resveratrol2 in several of the same cell lines (Huh-7, LNCaP, and PC-3), we were motivated to add to our dataset for these cells. Inclusion of SH-SY5Y cells was done to increase the breadth of cell type. Finally, an important consideration was that all four are widely used cell lines in cancer research and to study aspects of basic cell physiology and pathology (e.g., SH-SY5Y in neurodegenerative disease3 and ischemia/reperfusion models4 and Huh-7 to study hepatic xenobiotic metabolism5 and diseases such as NAFLD6). We have added several sentences in the introduction to provide our rationale.

2)      How did you arrive at 5% oxygen tension being physioxic for these cancers? 5% is physioxic for cartilage - https://www.frontiersin.org/articles/10.3389/fbioe.2020.590743/full but cartilage lacks blood vessels. A 2% level was found to be beneficial for adipose derived cells https://stemcellres.biomedcentral.com/articles/10.1186/s13287-018-0891-4 Even brief periods of atmospheric oxygen changed tumor biology vs. 3% oxygen tension https://www.science.org/doi/10.1126/sciadv.abh3375
Response: This is an excellent question, and indeed, physioxia is a range of O2 levels instead of a set value. Our choice of 5% was based on four considerations: (1) 5% O2 has been used in multiple studies using both non-transformed8–14 and cancer cells15–19; (2) we have previous data on metabolic and mitochondrial parameters from the same cell lines, using 5% and 18% O21; (3) 5% is admittedly at the higher end of the range for many cancer cells in vivo, but not by much, as values around 4% are also common20. (4) It is more difficult to regulate pericellular O2 levels in the media at levels below 5%. As the O2 gradient from incubator headspace to media diminishes, issues with hypoxia/anoxia around the cells increase. Using the same experimental conditions of this study, we have previously shown that at 5% O2 in the incubator headspace, pericellular O2 levels do not fall below 3.7%16. We are currently working on addressing this issue by designing an incubator that regulates the medium pericellular O2 level instead of headspace gas O2 level. To the best of our knowledge, at this time nobody is regulating, or routinely measuring, media pericellular O2 levels, so this is an important consideration that needs to be addressed next.

 We have added a couple sentences in the introduction to explain our choice.

3)      Was subculture etc. performed in a glove box (i.e. controlled oxygen environment) or in atmospheric conditions?
Response: This is an important question. While we unfortunately do not have access to a BSC that regulates O2 levels, we were careful to minimize the exposure to atmosphere during passaging. All media used during experiments is kept in the incubator at 5% O2 / 5% CO2 /37 deg C overnight to allow for gas equilibration. Nonetheless, media exchanges involve bringing these to the BSC and transferring. Hours are required for equilibration of O2 levels between gas and unstirred liquid phases21. Here, there is some physical disruption of media that allows mixing, but the transfers are completed in less than one minute, so any exposure to atmosphere is very brief. For subculturing using trypsinization, exposure to atmosphere is more extensive but still less than 15 min. We subculture every 3-4 days, depending on cell line. Cell harvesting for any endpoint measurements is always at least 3 days after seeding plates. Therefore, by the time of harvesting, the vast majority of cells was ‘born’ in the O2 condition of interest and would have experienced higher O2 levels for less than one minute each day.

4)      What cell passage was used in each case?

Response: Thank you for bringing this to our attention. In all cases, cell passages from 15-19 were used throughout this study. We have added a line in the methods section to clarify this.

5)      Line 84 – was overall transcript abundance the same, increased or decreased by oxygen tension?

Response: As stated in the second sentence of section 3.1, overall, a greater number of transcripts were overexpressed at 5% O2 than at 18% O2. In other words, the number of upregulated DEGs was greater at 5% than 18% O2.

6)      Fig 2 – some terms are lower case, others upper, please be consistent. Also, resolution of the words is low.

Response: Thank you for pointing this out. We have fixed these details with figure 2 by increasing the font size of the legends and axis titles in the figure. The resolution of the final figure is 300 dpi, adequate for publication.

7)      Figure and table legends indicate that the adjusted p value was not used as a cut off for inclusion. If the adjusted p value is not less than 0.05 it does not seem appropriate to include that data. This artificially inflates the number of differentially expressed genes.

Response: Many RNA-seq studies in the literature use raw p-value to threshold differential gene expression16,22–31. Because our main goal was to find patterns and trends in gene expression – such as the little overlap between DEGs, induction of HIF regulated genes, and differential regulation of mitochondrially-encoded genes – rather than focusing on specific gene targets, we believe that it is permissible to use a less stringent approach for cut-off, as has been done in the papers cited above. This way, we avoid the possibility of leaving out important genes (and/or pathways) that may have biological significance. In each instance (figures and tables) where we present DEGs and enrichment terms, we have been careful to clearly indicate which ones are below the threshold of adjusted p-value < 0.05 (and 0.005) and which are not, so that the reader can be properly informed of the statistical significance of the data.

References

  1. Moradi, F., Moffatt, C. & Stuart, J. A. The effect of oxygen and micronutrient composition of cell growth media on cancer cell bioenergetics and mitochondrial networks. Biomolecules 11, 1177 (2021).
  2. Fonseca, J., Moradi, F., Valente, A. J. F. & Stuart, J. A. Oxygen and glucose levels in cell culture media determine resveratrol’s effects on growth, hydrogen peroxide production, and mitochondrial dynamics. Antioxidants 7, 157 (2018).
  3. Xicoy, H., Wieringa, B. & Martens, G. J. M. The SH-SY5Y cell line in Parkinson’s disease research: a systematic review. Mol Neurodegener 12, 1–11 (2017).
  4. Liu, Y. et al. Human ischaemic cascade studies using SH-SY5Y cells: A systematic review and meta-analysis. Transl Stroke Res 9, 564–574 (2018).
  5. Bulutoglu, B. et al. A comparison of hepato-cellular in vitro platforms to study CYP3A4 induction. PLoS One 15, (2020).
  6. Müller, F. A. & Sturla, S. J. Human in vitro models of nonalcoholic fatty liver disease. Curr Opin Toxicol 16, 9–16 (2019).
  7. Keeley, T. P. & Mann, G. E. Defining Physiological Normoxia for Improved Translation of Cell Physiology to Animal Models and Humans. Physiol Rev 99, 161–234 (2019).
  8. Haas, B. et al. Permanent culture of macrophages at physiological oxygen attenuates the antioxidant and immunomodulatory properties of dimethyl fumarate. J Cell Physiol 230, 1128–1138 (2015).
  9. Zhu, J. et al. Physiological oxygen level is critical for modeling neuronal metabolism in vitro. J Neurosci Res 90, 422–434 (2012).
  10. Xie, P. et al. Physiological oxygen prevents frequent silencing of the DLK1-DIO3 cluster during human embryonic stem cells culture. Stem Cells 32, 391–401 (2014).
  11. Lees, J. G. et al. Oxygen regulates human pluripotent stem cell metabolic flux. Stem Cells Int 2019, 8195614 (2019).
  12. Keeley, T. P., Siow, R. C. M., Jacob, R. & Mann, G. E. A PP2A-mediated feedback mechanism controls Ca2+-dependent no synthesis under physiological oxygen. FASEB Journal 31, 5172–5183 (2017).
  13. Keeley, T. P., Siow, R. C. M., Jacob, R. & Mann, G. E. Reduced SERCA activity underlies dysregulation of Ca2+ homeostasis under atmospheric O2 levels. FASEB Journal 32, 2531–2538 (2018).
  14. Warpsinski, G. et al. Nrf2-regulated redox signaling in brain endothelial cells adapted to physiological oxygen levels: Consequences for sulforaphane mediated protection against hypoxia-reoxygenation. Redox Biol 37, (2020).
  15. Carrera, S. et al. Protection of cells in physiological oxygen tensions against DNA damage-induced apoptosis. Journal of Biological Chemistry 285, 13658–13665 (2010).
  16. Gardner, G. L. et al. Rapid nutrient depletion to below the physiological range by cancer cells cultured in Plasmax. Am J Physiol Cell Physiol 323, C823–C834 (2022).
  17. Maddalena, L. A. et al. Hydrogen peroxide production is affected by oxygen levels in mammalian cell culture. Biochem Biophys Res Commun 493, 246–251 (2017).
  18. Duś-Szachniewicz, K., Gdesz-Birula, K., Zduniak, K. & Wiśniewski, J. R. Proteomic-based analysis of hypoxia-and physioxia-responsive proteins and pathways in diffuse large B-cell lymphoma. Cells 10, (2021).
  19. Villeneuve, L., Tiede, L. M., Morsey, B. & Fox, H. S. Quantitative proteomics reveals oxygen-dependent changes in neuronal mitochondria affecting function and sensitivity to rotenone. J Proteome Res 12, 4599–4606 (2013).
  20. McKeown, S. R. Defining normoxia, physoxia and hypoxia in tumours - Implications for treatment response. British Journal of Radiology 87, 20130676 (2014).
  21. Bambrick, L. L., Kostov, Y. & Rao, G. In vitro cell culture pO2 is significantly different from incubator pO2. Biotechnol Prog 27, 1185–1189 (2011).
  22. He, K. et al. A transcriptomic study of myogenic differentiation under the overexpression of PPARγ by RNA-Seq. Sci Rep 7, (2017).
  23. Higareda-Almaraz, J. C. et al. Analysis and Prediction of Pathways in HeLa Cells by Integrating Biological Levels of Organization with Systems-Biology Approaches. PLoS One 8, (2013).
  24. Wen, Y. L. et al. Gene expression profiling and biofunction analysis of HepG2 cells targeted by crocetin. Mediators Inflamm 2021, (2021).
  25. Zhang, X., Gao, S., Li, Z., Wang, W. & Liu, G. Identification and Analysis of Estrogen Receptor Promoting Tamoxifen Resistance-Related lncRNAs. Biomed Res Int 2020, (2020).
  26. Zhou, R. et al. Salvianolic acid A attenuated myocardial infarction–induced apoptosis and inflammation by activating Trx. Naunyn Schmiedebergs Arch Pharmacol 393, 991–1002 (2020).
  27. Liu, S. B. et al. Histone methyltransferase KMT2D contributes to the protection of myocardial ischemic injury. Front Cell Dev Biol 10, (2022).
  28. Monroe, J. D. et al. RNA-Seq Analysis of Cisplatin and the Monofunctional Platinum(II) Complex, Phenanthriplatin, in A549 Non-Small Cell Lung Cancer and IMR90 Lung Fibroblast Cell Lines. Cells 9, (2020).
  29. Zhang, H. et al. Hypoxia regulates overall mRNA homeostasis by inducing Met1-linked linear ubiquitination of AGO2 in cancer cells. Nat Commun 12, (2021).
  30. Ma, H. S. et al. RNA-binding protein CELF6 modulates transcription and splicing levels of genes associated with tumorigenesis in lung cancer A549 cells. PeerJ 10, (2022).
  31. Li, Z. et al. DUS4L silencing suppresses cell proliferation and promotes apoptosis in human lung adenocarcinoma cell line A549. Cancer Manag Res 12, 9905–9913 (2020).

Reviewer 2 Report

In this work, the authors compare the changes in the transcriptome between physioxic (1-6%) and hyperoxic (18%) conditions. For this purpose, they use 4 different cancer cell lines, perform RNA-sequencing, identify differentially expressed genes (DEGs) between conditions, and explore functional significance. The conclusion from the study is that the oxygen concentration results in cell-type specific response and that the cell cultures must be performed in physioxic conditions. The article is well written and the results are clearly presented. However, in my opinion, the experiments and data presented are not sufficient to reach this conclusion. Most importantly, the conclusion that physioixia will increase the likelihood that results will translate to in vivo is not supported by this study.

The biggest issue with the study is the use of 4 different cancer cell lines originating from different tissue sources and comparing differences between them. Different response to stimuli can be expected from these cell lines given their origin, genetic background, and (past) culture conditions. The conclusions from the study would be better suited if the study was performed using two lines from the same tissue and comparing them to in vivo data. For instance, the authors have two prostate cancer lines available which could be compared to in vivo prostate cancer transcriptomic data. The issues with different sections of this manuscript are highlighted below.

1) Introduction

The introduction fails to link the goal of the study and the choice of studying this in cancer. Although this motivated later on, it should come in this section. It will clear to someone working in the cancer field but given the choice of journal, this must be stated. The choice of the cell lines is also not elaborated. It is also stated in the first line that the goal of mammalian cell culture is typically to model cell function in vivo. However, this is not an accurate statement and I would like to refer to the classification made by Russell and Burch on model systems. Model systems can be high fidelity or discrimination models. In the case of mammalian cell culture, their potential use is limited towards discrimination models that are suitable for studying one particular aspect of physiology or disease and are not high fidelity models aimed at modelling the whole in vivo cell function. Such expectations are unrealistic and unwarranted.

2) Methods

The methods section is missing some details. The cells are said to be cultured at 2.5% FBS. However, standard culture is performed at 10% FBS. Therefore, were the cells cultured at 10% FBS and switched to 2.5% FBS prior to the experiments? If this was the case, then how long were they given to acclimatize to the change of conditions?  Although it is clear to me what the DEGs refer to, it is not explicitly stated in the manuscript. For those unfamiliar with RNAseq this might not be apparent. It is also well known that these cell lines can be very different depending on how long they have been passaged and demonstrate clonal variability between passages. It is not stated for how long the authors cultured the cell lines before the experiment. It is also not stated if the authors performed any mycoplasma testing as mycoplasma contamination can drastically alter their transcriptome results.

3) Results

The results section starts immediately with the RNAseq results. However, some information on the viability of the different lines and changes in morphology, if any, would be an useful starting point towards understanding the behavior of different lines to the stimuli. In the results, it is stated that the O2 effects on gene expression are highly specific to a given cell line. Is this really surprising given the different origins of these cells? Additionally, these are all cancer cells which are aberrant and continue to diversify the longer they are kept in culture. Moreover, differences in response of different cell lines is to be expected. For instance, neurogenesis is seen in SH-SY5Y cell line. This is a neuroblastoma cell line while the others are not, so neurogenesis can be expected in the neuronal line but not the others.

4) Discussion

In the discussion section, certain aspects of the results are highlighted to claim that physioxic condition is more in vivo like. For different cell lines, based on their origin, one aspect is picked to show that physioxia is better but this is not thorough to make a conclusion as several other aspects may be worse. Thus, a broader characterization showing which pathways are better and which are worse is necessary before reaching conclusions. Moreover, the RNAseq data is used to indicate that the cell phenotype might be changing to an in vivo like situation. However, the transcriptomic changes does not mean there is changes at the proteomic level or that there are functional significance to their findings. The transcriptomic data should only serve as an indicator and further experiments must be performed to validate the finding. For instance, in the case of differentiation towards an osteoblast lineage, it must been shown with an assay that the cells are differentiating into osteoblasts. If there are changes in mitochondria numbers then it must be measured and quantified.

Based on my objections, the data presented here are merely indications and are not sufficient to reach conclusions. While the study has merit, it is necessary to perform more thorough experiments to support the conclusions reached by the authors.

Author Response

Reviewer 2

Comments and Suggestions for Authors

In this work, the authors compare the changes in the transcriptome between physioxic (1-6%) and hyperoxic (18%) conditions. For this purpose, they use 4 different cancer cell lines, perform RNA-sequencing, identify differentially expressed genes (DEGs) between conditions, and explore functional significance. The conclusion from the study is that the oxygen concentration results in cell-type specific response and that the cell cultures must be performed in physioxic conditions. The article is well written and the results are clearly presented. However, in my opinion, the experiments and data presented are not sufficient to reach this conclusion. Most importantly, the conclusion that physioixia will increase the likelihood that results will translate to in vivo is not supported by this study. 

The biggest issue with the study is the use of 4 different cancer cell lines originating from different tissue sources and comparing differences between them. Different response to stimuli can be expected from these cell lines given their origin, genetic background, and (past) culture conditions. The conclusions from the study would be better suited if the study was performed using two lines from the same tissue and comparing them to in vivo data. For instance, the authors have two prostate cancer lines available which could be compared to in vivo prostate cancer transcriptomic data. The issues with different sections of this manuscript are highlighted below.
Response: A handful of reports have investigated how culturing cell lines at physiological O2 levels induces gene expression patterns that better resemble primary tissues (see 1). However, this is actually not the specific question posed in our study. Here, we were interested to know how sensitive these cancer cells are to O2 levels in the range from physioxia (here 5%) to ‘normoxia’, i.e. 18% in the cell culture environment. This is directed more at determining whether it matters to incur the extra difficulty of regulating O2 during cell culture.

We did not necessarily ‘conclude’ that culturing cells in physioxia will increase the likelihood that results will translate to in vivo. It is more of an assumption based on the finding that we found a large transcriptomic signature of the elevated O2 level, which suggests that this deviation from the in vivo range of values will have a great effect. Generally, the further the cell culture model deviates from the in vivo condition, the greater the possibility of generating experimental artifact. This statement also applies to the use of physiologic media like HPLM or Plasmax2 as much as it would O2. We have tried to clarify this rationale by re-writing portions of the Introduction and Discussion.

1) Introduction
The introduction fails to link the goal of the study and the choice of studying this in cancer. Although this motivated later on, it should come in this section. It will clear to someone working in the cancer field but given the choice of journal, this must be stated. The choice of the cell lines is also not elaborated. It is also stated in the first line that the goal of mammalian cell culture is typically to model cell function in vivo. However, this is not an accurate statement and I would like to refer to the classification made by Russell and Burch on model systems. Model systems can be high fidelity or discrimination models. In the case of mammalian cell culture, their potential use is limited towards discrimination models that are suitable for studying one particular aspect of physiology or disease and are not high fidelity models aimed at modelling the whole in vivo cell function. Such expectations are unrealistic and unwarranted. 
Response: Thank you for this valuable observation. We have added several sentences to the introduction in order to explain the rationale for choosing cancer cells, and to explain the choice of cell lines. We have also rephrased the first sentence of the introduction to establish that studying certain aspects of cell biology in health and disease is one of the goals of mammalian cell culture, as suggested.

2) Methods
The methods section is missing some details. The cells are said to be cultured at 2.5% FBS. However, standard culture is performed at 10% FBS. Therefore, were the cells cultured at 10% FBS and switched to 2.5% FBS prior to the experiments? If this was the case, then how long were they given to acclimatize to the change of conditions?  Although it is clear to me what the DEGs refer to, it is not explicitly stated in the manuscript. For those unfamiliar with RNAseq this might not be apparent. It is also well known that these cell lines can be very different depending on how long they have been passaged and demonstrate clonal variability between passages. It is not stated for how long the authors cultured the cell lines before the experiment. It is also not stated if the authors performed any mycoplasma testing as mycoplasma contamination can drastically alter their transcriptome results.
Response: Thank you for pointing this out. We have modified section 2.1 of the methods to specify that cells were grown for a week to allow acclimatization to Plasmax and 2.5% FBS prior to exposing them to different O2 tension. We have also specified the number of passages used (15-19) and established that monitoring for mycoplasma contamination was performed routinely. We have also defined “DEGs” as “differentially expressed genes” in section 2.4.

3) Results
The results section starts immediately with the RNAseq results. However, some information on the viability of the different lines and changes in morphology, if any, would be an useful starting point towards understanding the behavior of different lines to the stimuli. In the results, it is stated that the O2 effects on gene expression are highly specific to a given cell line. Is this really surprising given the different origins of these cells? Additionally, these are all cancer cells which are aberrant and continue to diversify the longer they are kept in culture. Moreover, differences in response of different cell lines is to be expected. For instance, neurogenesis is seen in SH-SY5Y cell line. This is a neuroblastoma cell line while the others are not, so neurogenesis can be expected in the neuronal line but not the others.

Response: Multiple studies in the literature have already reported a number of functional and phenotypical changes in a variety of cell types (primary, immortalized, stem cells, differentiated cells), such as morphology, replicative lifespan and viability, cellular metabolism, and more. The goal of our study was to take a step back and investigate the breadth of the response to O2 levels between 5% O2 and 18% O2 by focusing on the transcriptome. Our approach also facilitated asking whether the response is similar or different across cell lines. With the little overlap between DEGs, differential regulation of HIF targets, and differential regulation of mitochondrially-encoded genes, we have demonstrated that the response to O2 in this range is indeed largely cell type-specific. This has an important message for experimentalists: it is not easy to predict how the hyperoxia of standard cell culture changes cell behavior in vitro, and how it might affect the outcome and validity of experiments. Incidentally, in the case of enrichment terms related to each cell line (e.g., neurogenesis in SH-SY5Y), this shows that indeed, O2 levels in this range may affect pathways that are highly relevant to the cell type being studied, further compromising the relevance of cell culture using supraphysiological O2 levels. We have rephrased the abstract, introduction, results, and discussion to emphasize the scope and limitations of this study.

4) Discussion
In the discussion section, certain aspects of the results are highlighted to claim that physioxic condition is more in vivo like. For different cell lines, based on their origin, one aspect is picked to show that physioxia is better but this is not thorough to make a conclusion as several other aspects may be worse. Thus, a broader characterization showing which pathways are better and which are worse is necessary before reaching conclusions. Moreover, the RNAseq data is used to indicate that the cell phenotype might be changing to an in vivo like situation. However, the transcriptomic changes does not mean there is changes at the proteomic level or that there are functional significance to their findings. The transcriptomic data should only serve as an indicator and further experiments must be performed to validate the finding. For instance, in the case of differentiation towards an osteoblast lineage, it must been shown with an assay that the cells are differentiating into osteoblasts. If there are changes in mitochondria numbers then it must be measured and quantified. 

Based on my objections, the data presented here are merely indications and are not sufficient to reach conclusions. While the study has merit, it is necessary to perform more thorough experiments to support the conclusions reached by the authors.

Response: The claim that physioxic conditions are more “in vivo like” is not a conclusion of this study, but rather its premise, which is based on the observation that O2 levels in routine cell culture are hyperoxic (~18% O2) compared to what most tissues encounter in physiological conditions in vivo (1-6%). As such, 5% O2 is indeed more “in vivo like” than 18% O2. A handful of reports has demonstrated how culturing cells in physioxia changes their phenotype in a way that better resembles the behavior of tissues in vivo1,3–5. Further, our aim was not to describe which pathways are “better” or “worse” under physioxia. The purpose of implementing physioxia is not to “improve” or “worsen” a pathological state or biological processes, but rather to try to more closely replicate the microenvironment of cells in vivo. Cell culturists routinely do this by maintaining temperature at 37 C degrees and pH at 7.

The purpose of functional enrichment analysis was to show which pathways, based on transcriptional patterns, may be affected by O2 in different cells. We recognize that changes in the transcriptome do not completely translate to changes in the proteome, or in the cell’s phenotype, but note that published papers have already noted specific phenotypic effects. One of the intentions of our study is to show trends and patterns in gene expression relating to exposure to the elevated O2 of standard cell culture. To address the concerns raised by the reviewer, we have rephrased the discussion to avoid over-concluding, and we have added an additional paragraph to explicitly establish the aims, scope, limitations, and findings of this study.

References

  1. Piossek, F. et al. Physiological oxygen and co-culture with human fibroblasts facilitate in vivo-like properties in human renal proximal tubular epithelial cells. Chem Biol Interact 361, (2022).
  2. vande Voorde, J. et al. Improving the metabolic fidelity of cancer models with a physiological cell culture medium. Sci Adv 5, eaau7314 (2019).
  3. Zhou, L., Dosanjh, A., Chen, H. & Karasek, M. Divergent Effects of Extracellular Oxygen on the Growth, Morphology, and Function of Human Skin Microvascular Endothelial Cells. J Cell Physiol 182, 134–140 (2000).
  4. Grodzki, A. C. G., Giulivi, C. & Lein, P. J. Oxygen Tension Modulates Differentiation and Primary Macrophage Functions in the Human Monocytic THP-1 Cell Line. PLoS One 8, (2013).
  5. Guo, R., Xu, X., Lu, Y. & Xie, X. Physiological oxygen tension reduces hepatocyte dedifferentiation in in vitro culture. Sci Rep 7, (2017).

Reviewer 3 Report

In this manuscript the authors investigated how supraphysiological O2 levels impact the cancer cell transcriptome compared to a more physiological O2 concentration (5%).

Overall data support the author’s conclusion. Only minor weaknesses should be addressed before publication: 

-Some acronyms, such as DEG or BMP should be better specified.

- Introduction: It would be useful to add some details on the rationale for choosing those 4 human cancer cell lines.

Author Response

Reviewer 3

In this manuscript the authors investigated how supraphysiological O2 levels impact the cancer cell transcriptome compared to a more physiological O2 concentration (5%).

Overall data support the author’s conclusion. Only minor weaknesses should be addressed before publication:

- Some acronyms, such as DEG or BMP should be better specified.
Response: Thank you for pointing this out. We have clearly defined these abbreviations on their first mention.

- Introduction: It would be useful to add some details on the rationale for choosing those 4 human cancer cell lines.
Response: Thank you for this comment. The questions we asked in this study were general and could have been addressed with any cell lines: firstly, how widespread are the effects of 18% O2, relative to physioxia, in cell culture? Secondly, are these effects common to all cell lines, or highly specific to an individual cell line? We have previously used three of the same cell lines (PC3, LNCaP, Huh7) to investigate the effects of 18% versus 5% O2 on specific cellular activities, like energy metabolism, mitochondrial network characteristics1 and response to drugs such as resveratrol2. Continuing with these same three cell lines for the present study therefore allowed us the possibility of integrating the transcriptome results into previous findings. Inclusion of SH-SY5Y cells was done to increase the breadth of cell types used. Finally, these are broadly used cell lines not only in cancer research, but also as surrogates of primary cells to study their physiology and pathology (e.g., SH-SY5Y in neurodegenerative disease3 and ischemia/reperfusion models4 and Huh-7 to study hepatic xenobiotic metabolism5 and diseases such as NAFLD6). We have added a couple of sentences in the introduction to provide our rationale.